# The Complex Interplay between the Gut Microbiome and Osteoarthritis: A Systematic Review on Potential Correlations and Therapeutic Approaches

**DOI:** 10.3390/ijms25010143

**Published:** 2023-12-21

**Authors:** Laura Marchese, Deyanira Contartese, Gianluca Giavaresi, Laura Di Sarno, Francesca Salamanna

**Affiliations:** Surgical Sciences and Technologies, IRCCS Istituto Ortopedico Rizzoli, Via di Barbiano 1/10, 40136 Bologna, Italy; laura.marchese@ior.it (L.M.); deyanira.contartese@ior.it (D.C.); laura.disarno@ior.it (L.D.S.); francesca.salamanna@ior.it (F.S.)

**Keywords:** osteoarthritis, gut microbiota, gastrointestinal microbiota, preclinical and clinical studies, systematic review

## Abstract

The objective of this review is to systematically analyze the potential correlation between gut microbiota and osteoarthritis (OA) as well as to evaluate the feasibility of microbiota-targeted therapies for treating OA. Studies conducted from October 2013 to October 2023 were identified via a search on electronic databases such as PubMed, Web of Science, and Scopus, following established PRISMA statement standards. Two reviewers independently screened, assessed, and extracted relevant data, and then they graded the studies using the ROBINS I tool for non-randomized interventions studies and SYRCLE’s risk-of-bias tool for animal studies. A search through 370 studies yielded 38 studies (24 preclinical and 14 clinical) that were included. In vivo research has predominantly concentrated on modifying the gut microbiota microenvironment, using dietary supplements, probiotics, and prebiotics to modify the OA status. *Lactobacilli* are the most thoroughly examined with *Lactobacillus acidophilus* found to effectively reduce cartilage damage, inflammatory factors, and pain. Additionally, *Lactobacillus M5* inhibits the development of OA by preventing high-fat diet (HFD)-induced obesity and protecting cartilage from damage. Although there are limited clinical studies, certain compositions of intestinal microbiota may be associated with onset and progression of OA, while others are linked to pain reduction in OA patients. Based on preclinical studies, there is evidence to suggest that the gut microbiota could play a significant role in the development and progression of OA. However, due to the scarcity of clinical studies, the exact mechanism linking the gut microbiota and OA remains unclear. Further research is necessary to evaluate specific gut microbiota compositions, potential pathogens, and their corresponding signaling pathways that contribute to the onset and progression of OA. This will help to validate the potential of targeting gut microbiota for treating OA patients.

## 1. Introduction

Osteoarthritis (OA) is a degenerative joint disease that progressively leads to the deterioration of articular cartilage [1]. With 300 million people affected worldwide, OA is a significant cause of pain and disability in adults. It is classified as the 15th most significant pathology contributing to years lived with disability [1].

Inflammation is widely recognized as a significant contributor to both the development and progression of OA. The most common inflammatory mediators associated with this condition include interleukins (ILs), namely IL6 and IL1β, metalloproteinases (MMPs), and tumor necrosis factor α (TNFα) [2]. Furthermore, the vascular endothelial growth factor (VEGF), the main stimulator of angiogenesis, and the activation of the innate immune system play a crucial role in the initiation and perpetuation of this inflammatory state. Angiogenesis is closely integrated processes in OA and may affect disease progression and pain. Furthermore, it has been reported that innate immunity plays a crucial role in the development of OA by activating synovial macrophages. This leads to the production of factors including ILs, which stimulate the production of MMPs and promote cartilage damage through aggrecans degradation. Thus, OA is the result of an imbalance between the anabolic and catabolic processes in the joint [3,4]. It is a multifactorial pathology and age, gender, genetics, obesity, metabolic syndrome, and endocrine factors can play a synergic role [5,6,7,8].

The diagnosis and treatment of OA typically involve a combination of clinical assessment, medical history, diagnostic tests, and various therapeutic approaches. Once diagnosed, the management of OA primarily involves a comprehensive approach aimed at relieving symptoms [9,10]. This approach includes physical therapy, drug therapy, and surgical intervention. Dieppe et al. proposed a management approach for osteoarthritis using a pyramid model [11]. The base includes education, lifestyle advice, exercise therapy, and topical non-steroidal anti-inflammatory drugs (NSAIDs). Progressing up the pyramid, oral NSAIDs, cyclooxygenase inhibitors, and orthoses are considered. If relief is insufficient, intraarticular injectables like corticosteroids may be used. Emerging therapies, such as stem cells or regenerative treatments like platelet-rich plasma (PRP), are gaining popularity, though they are not mentioned in the Dieppe pyramid [12,13]. However, none of these approaches has been shown to significantly modify disease progression. This is probably due to the numerous factors involved in the OA pathophysiology. Among these factors, the gut microbiota has gradually gained recognition as an important pathogenic factor in the development of OA [14,15,16,17], giving rise to the concept of the “gut–joint axis” that highlights the intricate interplay between gut microorganisms and joint health. Recently, two systematic reviews analyzed the crosstalk between OA and the intestinal microbiota, primarily focusing on gut microbiome composition, OA severity and pain, inflammatory factors, and intestinal permeability. However, none of these reviews explored the use of specific therapies for targeting the intestinal microbiota during OA. The intestinal microbiota comprises microorganisms that coexist symbiotically with the living organism in the gastrointestinal tract. It is mainly composed of bacteria but also includes viruses, fungi, archaea, and protozoa [18,19]. The physiological composition of the intestinal microbiota is primarily defined by *Firmicutes, Bacteroides, Proteus, Actinomycetes, Fusobacteria* and *Verruco microbia* with *Firmicutes* and *Bacteroides* being the dominant phyla (Figure 1) [18,19]. All these phyla can play critical roles in various physiological processes, including nutrient absorption, the maintenance of metabolic homeostasis, the development and maturation of the immune system, resistance to infections, protection against the development of systemic and mucosal immunity, and the production of neurotransmitters [18,19]. However, it is important to highlight that genetics background affects the environment where the microbiota lives, impacting the availability of nutrients and how the immune system works [20]. Host genetics significantly shape the microbiota, leading to different compositions among closely related species. Phylogenetically close species exhibit similar microbiota, and even within the same species, studies on twins show that monozygotic twins have a more similar microbiota than dizygotic twins, highlighting the crucial role genetics play in defining the microbiota [20].

Gut microbiota dysbiosis, defined as an alteration in the diversity, structure, or function of the intestinal microbiota, can contribute to the development of various pathological conditions and diseases [21,22]. Several factors and interventions can influence or regulate the microbiota population. Some keyways include dietary changes, probiotic supplementation, prebiotics, antibiotics, fecal microbiota transplantation, and lifestyle modifications. In the context of microbial dysbiosis and its association with inflammation, several pro-inflammatory cytokines may be involved, such as TNFα, IL1, IL6, IL17, and interferon-gamma [23]. These cytokines are part of the intricate network of immune signaling and can have both protective and detrimental effects depending on the context [23,24]. Dysregulation in their production, often triggered by microbial dysbiosis, is implicated in the pathogenesis of various inflammatory conditions and diseases, such as OA. In fact, the modulation of the microbiota has emerged as a potential factor in the effectiveness of certain anti-OA drugs [25]. While drugs like NSAIDs and corticosteroids may impact the gut microbiota, the direct influence of disease-modifying OA drugs (e.g., PRP) on the microbiota is not well established [25]. Additionally, diet and lifestyle interventions, known to influence the microbiota, may complement traditional OA treatments [26,27]. Several authors demonstrated that prebiotics, non-digestible dietary fibers, or compounds that serve as a food source for beneficial microorganisms in the gut can reverse the effect of a high-fat diet on OA by modulating the gut microbiota [26,27]. However, the interplay between the gut microbiota and anti-OA interventions is a growing area of research, highlighting the potential for optimizing OA management through microbiota modulation. To intervene in the gut–joint axis as part of treatment requires a more profound mechanistic understanding of microbiome–host interactions and a detailed characterization of the complex community interactions involved. Therefore, we conducted a systematic review to provide clarity and address specific questions: What are the specific correlations between OA and the gut microbiota? What are the potentials of microbiota-targeted therapies used during OA? This systematic review evaluated the potential mechanisms of the association between gut microbiota and OA and tried to assess the potentials of microbiota-targeted therapies in OA.

## 2. Methods

### 2.1. Eligibility Criteria

The PICOS (Population, Intervention, Comparison, Outcomes, Study design) model was used to conduct this review. Studies that evaluated the relationship between gut microbiota and OA in patients and animals (Population), with or without a specific intervention/treatment (Intervention), with or without a comparison group (Comparison), and that described the relationships between gut microbiota and OA (Outcomes) in preclinical and clinical studies (Study design) were included. Studies from October 2013 to October 2023, were included in this review if they met the PICOS criteria. Articles that focused on OA but did not discuss gut microbiota, and vice versa, articles that analyzed the presence of other pathological conditions beyond OA, articles that did not specified intervention or specific therapies used and articles that did not assess the relationship between OA and gut microbiota were excluded. We also excluded studies in which data were not accessible or missing or those without an available full-text article. Furthermore, duplicates, reviews, letters, comment to the editor, protocols and recommendations, editorials, guidelines, and articles not written in English were excluded.

### 2.2. Search Strategies

A systematic literature review was conducted in October 2023 according to the Preferred Reporting Items for Systematic Reviews and Meta-Analyses (PRISMA) [28] (Appendix A). The search was conducted on three databases: PubMed, Scopus, and Web of Science. The resulting combination of terms was used: “osteoarthritis” OR “osteoarthritides” AND “gastrointestinal microbiome” OR “gut microbiota”. For each term, free words, and managed vocabulary specific to each bibliographic database were merged using the operator “OR”. The combination of free vocabulary and/or Medical Subject Headings (MeSH) terms were reported in Table 1.

### 2.3. Selection Process

Following the removal of duplicate articles, using a public reference manager (Mendeley Desktop v.1.19.8), potentially relevant articles were initially screened by two reviewers (LM, FS) based on their titles and abstracts. Any studies that did not meet the predefined inclusion criteria were excluded, and any uncertainties were addressed by involving a third reviewer (GG). Finally, the remaining studies were included in the final stage of data extraction.

### 2.4. Data Collection Process and Synthesis Methods

The process of data extraction and synthesis started with a systematic cataloguing of the details contained in the studies under review. To increase the validity of the process and to ensure that potentially relevant findings were not inadvertently overlooked during synthesis, two authors (LM, FS) undertook the extraction task. This involved the preparation of 2 tables (one for preclinical studies and one for clinical studies), in which various key elements were carefully recorded. The elements in Table 2 relating to preclinical studies included the following categories: species/age/sex/animals’ number, OA model, aim, treatment, experimental groups and time, OA assessment, microbiome assessment, main results, relation between OA and gut dysbiosis, reference/year/country, and Systematic Review Centre for Laboratory Animal Experimentation (SYRCLE) risk of bias [29]. Instead, the data collected in Table 3, which focused on clinical studies, included the following aspects: study design, age/gender/number, OA assessment methods, study aim, treatment, groups, follow-up or experimental time, microbiota assessment, main results, ref./year/country, ROBINS-I risk of bias [30].

### 2.5. Risk of Bias Assessment

Two reviewers (LM and DC) analyzed the methodological quality of the included studies individually. In case of disagreement, they tried to reach consensus; if this failed, a third reviewer (FS) made the final decision. The methodological quality of the included in vivo studies was ensured according to the SYRCLE tool [29], which assesses the risk of bias of animal studies. The methodological quality of the included clinical studies was assessed using the ROBINS-I tool for the assessment of risk of bias in non-randomized studies of interventions [30].

## 3. Results

### 3.1. Study Selection

The initial search found 370 studies. Of these, 123 were found using PubMed, 74 were found using Scopus, and 173 were found in Web of Science. Articles were uploaded to Mendeley Desktop version 1.17.9 to remove duplicates, and the resulting 78 articles were screened for title and abstract. Seventy-eight complete articles were screened to determine whether the publication met the inclusion criteria, and 38 were considered eligible for the review. Of the 38 articles eligible for the review, 24 were in vivo studies and 14 were clinical studies. The search strategy and study inclusion and exclusion criteria are shown in Figure 2.

The review was registered to OSF registers (DOI 10.17605/OSF.IO/5JUQC).

### 3.2. General Characteristics of In Vivo Studies

Of the 38 studies identified in this review, 24 were preclinical in vivo studies (Table 2). Of these, 11 used an OA rat animal model (5 chemically induced OA [31,32,37,41,43], 1 surgically induced OA [36], 3 high fat diet-induced OA [34,42,44], 1 deoxycorticosterone acetate (DOCA) induced [46] and 1 antibiotic + tryptophan rich diet induced [40]), 11 used an OA mouse model (7 surgically induced OA [26,27,33,35,39,50,51], 2 high fat diet induced [44,45], 2 load induced [39,49], 1 used a spontaneous OA model in guinea pigs [52]) and 1 used a spontaneous OA model in rhesus macaques [47]. The number of animals used in these studies was variable (range: from 20 to 88) and not reported in a few articles (*n* = 3) [26,31,38]. The main technique used to assess OA was histology, which in some studies was also combined with specific semi-quantitative scores (Mankin’s and Osteoarthritis Research Society International (OARSI)) [26,33,37,39,40,50] and/or immunohistochemistry [33,46], ramp test [32], micro-computed tomography [27,32,46,49,51], anterior drawer test, Magnetic Resonance Imaging [47], and dual-energy X-ray absorptiometry [27]. Some articles (*n* = 5) were designed to test the changes in the gut microbiota and metabolism during the administration of various supplements, such as quercetin, bioactive compounds from chicken cartilage, extracellular vesicles (EVs) from raw milk, oligofructose and prebiotic fiber, for OA [26,31,32,33,34]. Four other studies investigated the impact of probiotic therapy on OA and its relationship with gut microbiota. Specifically, one study tested Lactobacillus Acidophilus [37], one studied Lactobacillus Acidophilus in combination with Bifidobacterium breve [35], one evaluated Clostridium Butyricum [36] and one tested a probiotic combination (Lactobacillus paracasei subsp. paracasei M5 and Costridium Sulfate) [38]. Three other studies investigated the association between gut dysbiosis induced by antibiotic administration (ampicillin, neomycin or antibiotic combined with a tryptophan-rich diet) and OA [27,39,40]. The effect of traditional Chinese medicine therapies, such as moxibustion and eletroacupuncture (EA), on OA and gut microbiota was reviewed in three articles [41,42,43]. The effect of high-fat diet (HFD) on OA and the gut microbiome was also investigated by Collins et al. and by Li et al. [44,45]. HFD leads to weight gain, which results in increased mechanical stress on the joints, systemic inflammation, glucose intolerance, and an increase in local leptin signaling. One additional study [46] evaluated OA and gut microbiome alteration in a DOCA-induced hypertensive model. Finally, six studies evaluated the relationship between OA and gut microbiota composition without specific treatments and/or procedures.

In all the analyzed studies, the gut microbiota was detected by 16S rRNA amplification, which is a DNA sequence-based method that can identify different bacteria.

#### In Vivo Studies Results

Using different animal models of OA, studies evaluating oral supplementation during OA reported specific changes in the gut microbiota and metabolism. Specifically, in animals with chemically induced OA, the intragastric administration of quercetin led to a decrease in short-chain fatty acids and an increase in Lactobacillus and Ruminococcacea [31]. Using the same OA animal model, it was demonstrated that oral bioactive compounds from chicken cartilage, specifically chondroitin sulfate and type II collagen peptides, were associated with a decrease in inflammatory cytokines. In a surgical OA model, EVs supplementation led to a reduction in senescent cells [33]. Meanwhile, in a metabolic OA model, probiotic fiber supplementation reduced joint damage, leptin levels, lipid endotoxins, and alleviated dysbiosis [34]. In the same animal model, oligofructose supplementation demonstrated a protective role against trauma-induced obesity-associated OA, which was primarily due to its effect on the gut microbiome [45]. Similarly, in studies evaluating the effect of probiotic administration by using surgically induced OA model [35] and a chemically induced OA model [37], researchers found that Lactobacillus Acidophilus led to a reduction in inflammatory factors, cartilage damage, and pain [35,37]. Beyond local effects in joint tissues, O’Sullivan et al. also observed the suppression of angiogenesis factors VEGF-A and VEGF-R1 in the distal colon combined with reduced inflammatory cytokines in peripheral tissues of Lactobacillus Acidophilus-treated OA mice. This indicates a close connection between the oral administration of Lactobacillus Acidophilus and its systemic impact throughout the body, affecting joint tissue homeostasis [35]. In addition, Clostridium Butyricum administration reduced OA damage in one study [36]. Lactobacillus M5 also inhibited the development of OA by preventing the development of HFD-induced obesity, protecting cartilage from damage, regulating adipokine levels, and modulating the composition of the gut microbiota in mice [38].

The studies investigating the association between gut dysbiosis induced by antibiotic administration (ampicillin, neomycin or antibiotic combined with a tryptophan-rich diet) and OA [27,39,40], using surgically induced or HFD OA models, showed that antibiotic-induced gut dysbiosis reduced the level of lipopolysaccharide (LPS) and the inflammatory responses. They have also shown that antibiotic-induced gut microbiota dysbiosis reduces serum LPS levels and inflammatory responses, including the suppression of TNF-α and IL-6 levels. These changes may lead to a decrease in MMP-13 expression and an improvement in OA following joint injury [27,39,40]. Thus, the depletion of the gut flora by antibiotics reduced the microbial products that suppress pro-inflammatory cytokines, thereby delaying the development of OA. Furthermore, it was demonstrated that adverse alterations in the intestinal microbiome, particularly in tryptophan metabolism, accelerated the development of OA through its interaction with aryl hydrocarbon receptor [40]. Similarly, studies of traditional Chinese medicine therapies showed that maxibustion regulates the composition of the gut microbiota, leading to a reduction in OA cartilage damage. In addition, EA treatment of ST36 and GB34 prevents OA degradation by modulating lipid metabolism and gut microbiota [41,42,43].

A study by Chan et al. demonstrated that DOCA-induced hypertension led to the accumulation of p16INK4a+ senescent cells (SnCs) in the knee joint. This accumulation contributed to OA development [46]. Captopril, an anti-hypertensive drug, was effective in removing p16INK4a+ SnCs and reducing OA damage [46]. Furthermore, these changes were associated with a reduction in Escherichia–Shigella levels in the gut microbiome. Therefore, gut microbiota dysbiosis emerged as a metabolic link in chondrocyte senescence induced by DOCA-triggered hypertension [46].

The studies that evaluated the association between gut microbiota and OA, without any treatment and/or by evaluating the effect of HFD, found an association between OA and gut microbiota [43,44,45,47,48,49,50,51,52]. A greater richness in gut microbiota composition was positively associated with a reduction in OA. Lactobacillus spp. was positively correlated with OA in one study and negatively correlated in another [35,44]. *Metanobrevibacter* spp. was correlated with a higher OA score. Prevotella and Ruminococcus were negatively associated with OA in one study [36].

### 3.3. General Characteristics of Clinical Studies

Of the 38 studies included in this review, 14 were clinical studies (Table 3). Among these clinical studies, there was variability in the number of patients involved with the majority having an average of 75 patients. Only 3 studies had a different number of patients, with 2 having more than 10,000 patients [56] and 1 having only one patient [54]. The age of the patients in the studies varied and ranged from a minimum of 18 years [54] to a maximum of 75 years [55]. In seven studies, the anatomic site of OA was localized in the knee. In four studies, OA was localized in the knee with either the hand or the hip, while in four studies, the OA site was left unspecified, or multiple sites were delineated. The methods used to detect OA were X-ray combined with Kellgren and Lawrence score or WOMAC score.

#### Clinical Studies Results

Almost all the studies (13/14) aimed to establish a link between OA and gut microbiota composition, and several associations were identified [53,54,55,56,57,58,59,60,61,62,63,64,65]. Gut microbiota composition was found to be altered in OA patients compared to healthy controls, involving changes both in composition and functionality as well as chronic low-grade inflammation. The Methanobacteriaceae family, Desulfoovibrionales order and Ruminiclostridiu 5 genus were negatively associated with OA, having a protective effect against the disease [56]. Similarly, also Bacteroides, Agathobacter, Faecalli bacterium, and Roseburia showed a protective effect while Streptococcus and Enterococcus were associated with increased pain during OA. In addition, greater levels of Streptococcus spp. were significantly associated with higher WOMAC scores independently to possible confounders such as smoking, alcohol consumption and body mass index [57]. Higher levels of Clostridiales and Firmicutes were also found in OA patients. Chen et al. also showed that OA patients can be discriminated from healthy controls using the multi-kingdom signatures, an analysis that identifies interactions between different bacteria, suggesting the potential of gut microbiota for predicting OA [62].

Two studies looked at the effect of probiotics on OA, and both showed a reduction in pain [27,54]. In addition, one study tested the effect of EA on OA and found that EA increase Bacteroides and Agathobacter, contributing to the disease improvement [57]. This increase may also offer protection against inflammation. In three studies [53,64,66], a direct relationship between OA and gut microbiota composition was not found. However, in two of these, one cross-sectional and one prospective, an increase in LPS was found in OA patients. This implies an indirect connection between OA and the gut microbiota given that LPS is generated by Gram-negative bacteria in the intestinal tract [65].

### 3.4. Risk of Bias Assessment

For the in vivo studies (*n* = 24), the risk of bias, reported in Figure 3, was high in almost all the studies. Approximately 72% of the studies did not report the method of sequence generation (*n* = 18). Almost all the studies (92%) reported that the groups were similar in terms of baseline characteristics (*n* = 23). The allocation was not adequately concealed in about 80% of the studies (*n* = 20), while it was concealed in four studies [32,33,40,45]. Only one study reported that the animals were randomized during the experiment [49], while three studies used the housed blinding [40,43,66]. One study selected assessors blinding [36], and two studies employed random outcome assessment [43,66]. All studies included all the animals in the analyses (*n* = 24), reported the primary outcomes (*n* = 24) and were free of other biases that could lead to a high risk (*n* = 24).

The assessment of risk of bias for the fourteen clinical trials included in this review is shown in Figure 4. For these studies, the risk of bias was mainly low with only one domain in some studies presenting a moderate risk, i.e., the selection of participants into the study in pre-intervention [53], missing data in post-intervention [53,54,57], measurement of outcomes in post-intervention [63,66], and selection of the reported result in post-intervention [57,59,61,63,64].

## 4. Discussion

This review presents evidence from in vivo studies demonstrating that nutrients in the diet and prebiotics can improve the status of OA by modulating the microenvironment of the gut microbiota, including changes in composition and metabolism. This leads to a reduction in joint damage, senescent cells, leptin levels, lipids, endotoxins, inflammatory factors and pain levels. Among probiotics, *Lactobacillus* is certainly the most discussed in these papers [35,37,38,54,66]. Preclinical data revealed that specifically the levels of *Lactobacillus* and other bacterial species (e.g., *Bifidobacterium, Clostridium, Streptococcus, Bacteroides and Firmicutes*) have the potential to be a crucial factor in the development of OA. The specific mechanisms that underlie this relationship remain under investigation, but the hypothesis is that the gut microbiome could impact systemic inflammation and immune responses, subsequently affecting joint health [37]. In this review, only one study [47] reported an opposite correlation, where *Lactobacillus* levels were increased in OA.

Some preclinical studies, within the context of OA, also investigated antibiotics, which can alter the gut bacteria composition, in addition to dietary nutrients and prebiotics. These studies have investigated how changes in the gut microbiome induced by antibiotics might influence OA symptoms. Specifically, some research has indicated that antibiotic-induced modifications in the gut microbiome could be associated with a reduction in OA symptoms [27,39,40]. However, it is likely that the effects of antibiotics on OA may vary depending on individual factors, including the specific antibiotics used and the unique composition of the gut microbiome. Additional studies are necessary to gain a better understanding of the precise mechanisms by which the gut microbiome and antibiotics may impact OA. This could pave the way for more targeted approaches to managing OA through interventions in the gut microbiome.

Several preclinical studies also analyzed the relationship between HFD, OA, and the gut microbiota [42,44,45]. HFD can lead to changes in the composition of the gut microbiota. Such diets can alter the balance of bacteria in the gut, potentially promoting the growth of pro-inflammatory microbes while reducing beneficial ones [44,45]; these changes in gut microbiota composition were linked to systemic inflammation and metabolic disturbances, which are associated with OA. Although animal studies have indicated a relationship between diet, gut microbiota and OA-related factors, research on humans is sparse. Only one study has investigated the translation of these findings, which did not show disparity in the fecal microbial communities between obese adults with OA and obese controls without OA [61].

In addition to the preclinical data, some clinical studies demonstrated that *Lactobacillus* leads to a decrease in various inflammatory factors as well as nociceptive mediators [27,54]. Furthermore, a correlation was also demonstrated between elevated levels of *Lactobacillus* and a decrease in OA symptoms. Gut microbiota alterations, such as increased *Clostridiales* and *Firmicutes* levels, were also observed in OA patients. Furthermore, some specific bacteria were associated with lower risk of OA (*Bifidobacterium*), while others were linked to higher pain and disability scores in OA patients (*Clostridium, Streptococcus, Bacteroides and Firmicutes*). Despite these interesting findings, future larger clinical studies are mandatory to strengthen these concepts. This aspect was further confirmed by the search on ClinicalTrials.gov for ongoing clinical studies on “osteoarthritis” and “intestinal microbiome”. The search yielded only four clinical studies (NCT05186714, NCT03985709, NCT03968770, NCT04172688). One of these studies is in the recruitment phase (NCT05186714), two have an unknown status (NCT03985709, NCT03968770), and the last one is completed, but no results have been posted (NCT04172688). In detail, one clinical study (NCT05186714) uses 16S rRNA amplicon sequencing of fecal DNA samples to compare changes in gut microbiota profiles and the quantity of Streptococcus species with OA pain; the second study investigates the association between changes in gut microbiota and the symptoms of knee and/or hip OA in Italian patients (NCT03985709). Finally, two studies evaluated the association between probiotics and gut microbiota and the intensity of OA pain (NCT03968770, NCT04172688).

A limitation of this systematic review is its descriptive approach. No meta-analysis of the included articles was performed due to the presence of statistically significant heterogeneity between them. This aspect was particularly evident in preclinical studies, where the risk of bias was high in almost all of them. In contrast, clinical data were generally of moderate to high quality and indicated a consistent correlation between OA and specific bacterial strains and phyla ratio in the gut microbiome. However, it should be noted that only a few of them have been reported, and none of them are large. To intervene in the gut–joint axis as part of treatment, a deeper mechanistic understanding of microbiome–host interactions and a detailed characterization of the intricate community interactions involved are necessary. Nevertheless, this review presents a thorough examination of the existing knowledge on the connection between gut microbiota and OA. It identifies specific areas that require further investigation: (1) exploring the mechanisms linking the gut microbiota and OA; (2) examining the shared pathways and synergies between probiotics, antibiotics, diet, and nutraceuticals in regulating the gut microbiota and OA; (3) investigating the relationship between age, gender, and OA-associated dysbiosis; (4) investigating the quantitative relationship between OA and the gut microbiota; and (5) investigating the relationship between dysbiosis, OA disease, and symptoms.

## 5. Conclusions

In this review, we have comprehensively examined the connection between gut microbiota and the progression of OA as well as the potential for altering gut microbiota in OA treatment, providing evidence for the existence of a gut–joint axis in the OA pathogenesis. Both preclinical and clinical studies provide evidence suggesting that probiotics have the potential to be advantageous for patients experiencing OA pain. This is achieved through the positive modulation of gut microbiota and the attenuation of low-grade inflammation via multiple pathways, as indicated by a growing body of research particularly in the laboratory settings. Furthermore, there is a pressing need for a more comprehensive exploration of confounding factors, particularly genetic background, sex, age, and socio-economic context. This exploration should encompass levels of physical activity, dietary composition, and the use of concomitant prescribed medications. In addition to probiotics, also antibiotics, dietary interventions, and nutraceuticals demonstrated a role in regulating gut microbiota. However, given these potentially beneficial effects among antibiotics, dietary interventions, and nutraceuticals, it is imperative that further and larger-scale future studies be conducted to delve deeper into gut microbiome diversity also through next-generation sequencing, transcriptomics, and metabolomics. These discoveries may pave the way for OA treatment through targeted interventions in the gut microbiota.

## Figures and Tables

**Figure 1 ijms-25-00143-f001:**
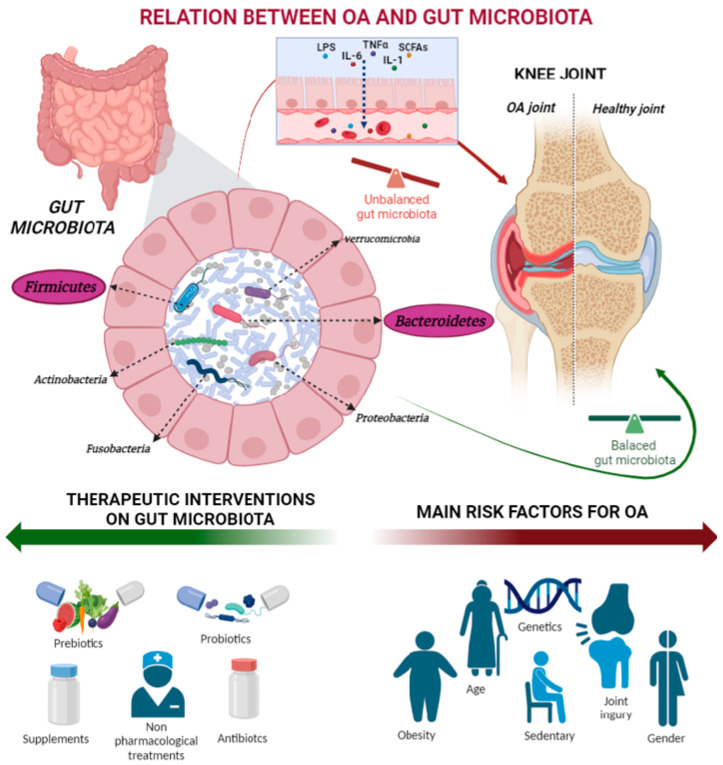
Schematic representation of the interaction between gut microbiota and OA. On the left side: gut microbiota representation with the main phylum expressed, *Firmicutes*, *Actinobacteria*, *Fusobacteria*, *Proteobacteria*, *Bacteroidetes*, *Verrucomicrobia*; *Firmicutes* and *Bacteroidetes* were the most expressed. Right side: OA and healthy knee joint. The green arrow represents a good gut microbiota condition, and it is correlated with healthy cartilage, while the red arrow represents an imbalanced gut microbiota, and it is correlated with OA.

**Figure 2 ijms-25-00143-f002:**
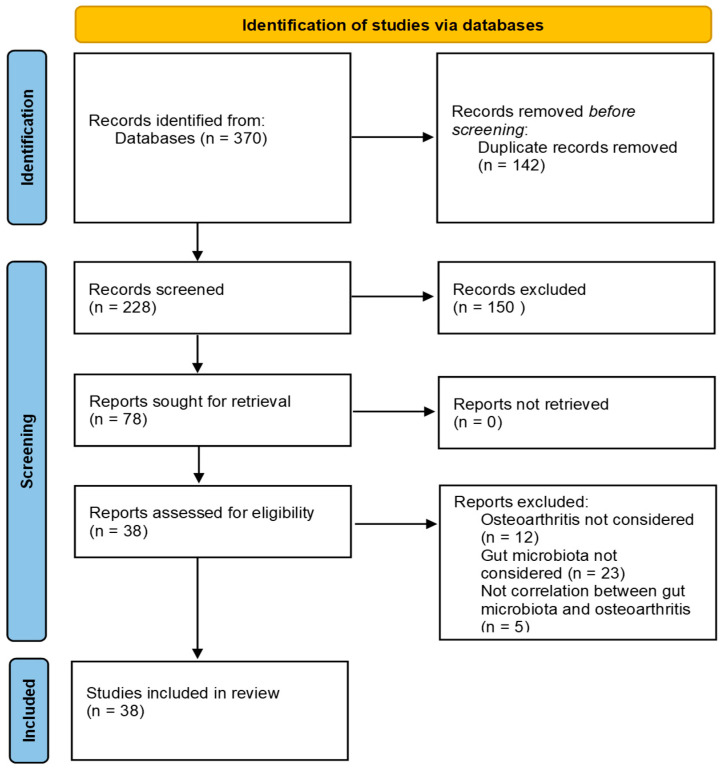
The PRISMA flow diagram 2020 for the systematic review.

**Figure 3 ijms-25-00143-f003:**
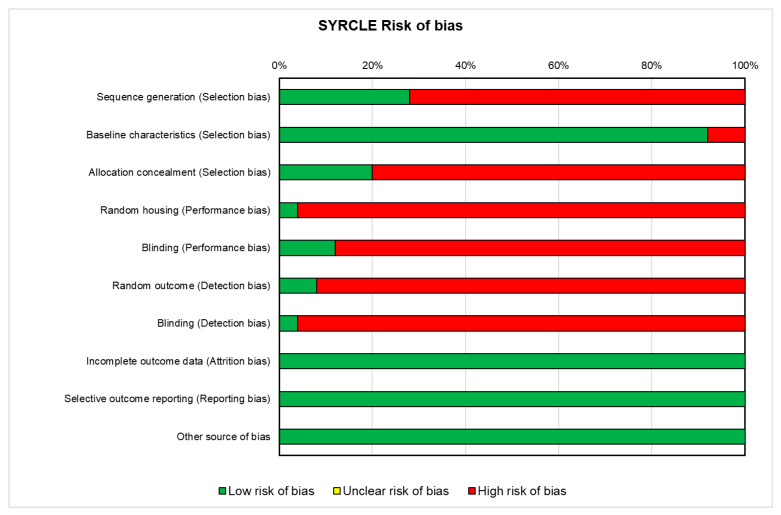
SYRCLE’s tool for assessing risk of bias in the in vivo studies. Green: low risk of bias; Red: high risk of bias. (1) Low risk of bias (the study is comparable to a well-performed randomized study); (2) High risk of bias (the study has some important problems). Details are reported in the Appendix A.

**Figure 4 ijms-25-00143-f004:**
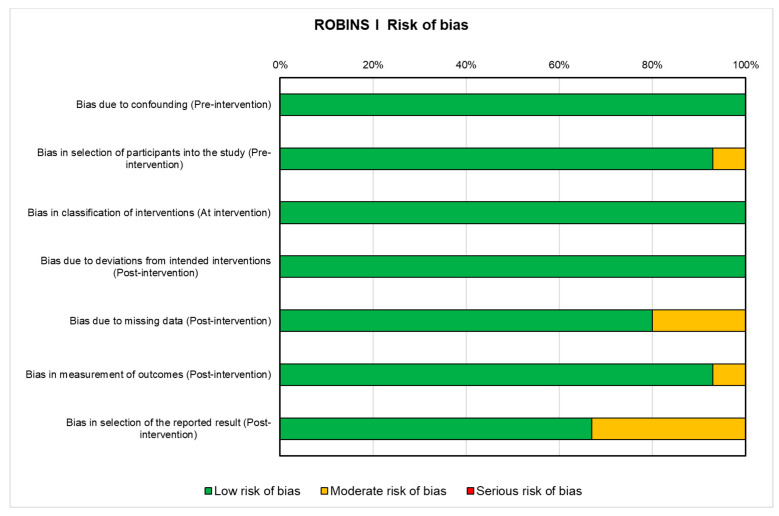
ROBINS-I tool for assessing risk of bias in non-randomized clinical studies. Green: low risk of bias; Yellow: moderate risk of bias. (1) Low risk of bias (the study is comparable to a well-performed randomized trial); (2) moderate risk of bias (the study provides sound evidence for a non-randomized study but cannot be considered comparable to a well-performed randomized trial). Details are reported in the Appendix A.

**Table 1 ijms-25-00143-t001:** Combination of free-vocabulary and/or MeSH terms for the identification of studies in PubMed, Scopus and Web of Science.

Database	MeSH Terms
PubMed	((“osteoarthritis”[MeSH Terms] OR “osteoarthritis”[All Fields] OR “osteoarthritides”[All Fields]) AND (“gastrointestinal microbiome”[MeSH Terms] OR (“gastrointestinal”[All Fields] AND “microbiome”[All Fields]) OR “gastrointestinal microbiome”[All Fields] OR (“gut”[All Fields] AND “microbiota”[All Fields]) OR “gut microbiota”[All Fields])) AND (y_10[Filter])
Scopus	TITLE-ABS-KEY TITLE-ABS-KEY (osteoarthritis)) AND ((TITLE-ABS-KEY (gut AND microbiota)) OR (TITLE-ABS-KEY (gastrointestinal AND microbiome))) AND PUBYEAR > 2012 AND (LIMIT-TO (LANGUAGE, “English”)) AND (LIM-IT-TO (DOCTYPE, “ar”))
Web of Science	(TS = osteoarthritis OR TS = osteoarthritides) AND (TS = gut microbiota OR TS = gastrointestinal microbiome)—with Publication Year from 2013 to 2023, English

**Table 2 ijms-25-00143-t002:** Main characteristics of preclinical studies included in the review.

Treatment Classification	Species/Age/Sex/ Animals Number	OA Model	Aim	Treatment	Experimental Groups and Time	OA Assessment	Microbiome Assessment	Main Results	Relation between OA and Gut Dysbiosis	Ref. Year, Country	SYRCLE Risk of Bias
Supplements	Rats, NR, NR, NR	MIA	To evaluate changes in gut microbiota and metabolism during quercetin therapy for OA	Quercetin	(1) OA; (2) Quercetin; (3) Control 29 days	NR	DNA sequencing (targeting 16s rRNA gene)	↓SCFAs (acetic and propionic acid) in OA group vs. control; ↑SCFAs (acetic and propionic acid) in quercetin group vs. OA group; ↑*LA* and ↓*Ruminococcacea* in quercetin group vs. all groups	Quercetin treatment influences the features and composition of gut microbiota as well as metabolism in OA	Lan et al. 2021 [31], USA	High
Supplements	SD rats, 8–9 weeks old, male, 30	MIA	To evaluate the effects of bioactive compounds of chicken cartilage for OA treatment	Chicken cartilage food supplements	(1) Sham (oral gavage of saline); (2) OA + oral gavage of saline; (3) OA + oral gavage of CS; (4) OA + oral gavage of collagen peptides; (5) OA + oral gavage of diacerein 4 weeks	Rotating rod test, ramp test, histopathology, Mankin score	DNA sequencing (targeting V3–V4 region of 16s rRNA gene)	↑Athletic ability and ↓IL1β, IL6, PGE2, TNFα with CS treatment	CS reduces OA development by modulating gut microbiota	Zhang et al. 2022 [32], China	Moderate
Supplements	C57BL/6J mice, male, 6 weeks old, 57	DMM	The therapeutic potential of mEVs for OA treatment	EV isolated from raw milk	(1) Healthy control; (2) DMM + PBS; (3) DMM + mEVs 10 weeks	Serum ELISA, histology, OARSI score, IHC	DNA sequencing (targeting V3–V4 region of 16s rRNA gene)	↓MMP-1, diversity of gut microbiota and ↑COL2A1 in DMM + mEVs group vs. DMM + PBS group	mEV reduces cartilage degradation by decreasing catabolic protein expression and restoring the gut microbiota	Liu et al. 2023 [33], China	Moderate
Supplements	SD rats, 12 weeks old, NR, 49	HFS diet	To determine the effects of prebiotic fiber supplementation, aerobic exercise, and the combination of the two, on OA development	Prebiotic fiber	(1) Sedentary + standard chow diet; (2) Sedentary + HFS diet; (3) Non-exercise + HFS diet + prebiotic fiber; (4) Exercise + HFS diet; (5) Exercise + HFS diet + prebiotic fiber 12 weeks	Histology	DNA sequencing (targeting 16s rRNA gene)	↓Joint damage, leptin level, lipid profile, endotoxin in non-exercise + HFS diet + prebiotic fiber, exercise + HFS diet and exercise + HFS diet + prebiotic fiber groups; ↓dysbiosis in non-exercise + HFS diet + prebiotic fiber group	Prebiotic fiber supplementation, aerobic exercise, and the combination of both treatments prevent OA also by influencing inflammation and dysbiosis	Rios et al. 2019 [34], Canada	Moderate
Supplements	C57BL/6J mice, 5 weeks old, NR, NR	DMM	To evaluate the impact of obesity and oligofructose on gut microbiome and OA	Oligofructose supplementation	Low or HFD + diet supplemented with control fiber or prebiotic + DMM or sham surgery 2 weeks after DMM	Histology, OARSI score	DNA sequencing (targeting V3–V4 region of 16s rRNA gene)	Oligofructose treatment ↑colon transcriptome, ↓colonic macrophage cell signature and joint inflammation, preserving articular cartilage and protecting against OA	Oligofructose supplementation impacts on the gut microbiome and plays a protective role against trauma-induced OA	Schott et al. 2018 [26], USA	High
Probiotics	C57BL/6 mice, 11 weeks old, female, 40	PMM	To evaluate the *LA1* effect OA	*LA1*, *Bifidobacterium breve*, *LA reuteri*, *Bacillus subtilis*	(1) OA + PBS; (2) OA + *LA1*; (3) OA + *Bifidobacterium breve* (4) OA + *LA reuteri*; (5) OA + *Bacillus subtilis* 9 and 12 weeks after surgery	Histology	DNA sequencing (targeting 16s rRNA gene)	↓IL1β, TNFα, NFKB, NLRP3, VEGF, NGF, artemin, GFRα3, RUNX2, MMP13, pain, heat tolerance and cartilage damage and ↑genus *Akkermanis munniphila* and *Lachnospiraceae* with *LA1* treatment	*LA1* protect joint tissue integrity during OA progression, as shown by significant alterations in the OA gut microbiome	O-Sullivan et al. 2022 [35], USA	High
Probiotics	Wistar rats, NR, female, 30	ACL dissection	To investigate the effects of MY, a product made from *Clostridium butyricum*, on OA	MY	(1) Control; (2) OA; (3) OA + MY 4 weeks	Histology	DNA sequencing (targeting 16s rRNA gene)	MY ↑Chao1, Shannon and Pielou vs. untreated group; ↑*Prevotella*, *Ruminococcus*, *Desulfovibrio*, *Shigella*, *Helicobacter* and *Streptococcus* in OA group; ↑*LA*, *Oscillospira*, *Clostridium* and *Coprococcus* with MY treatment	MY protect against OA via the gut–muscle–joint axis by increasing the beneficial bacteria	Xu et al. 2022 [36], China	High
Probiotics	Wistar rats, 7 weeks old, male, 6	MIA	Investigate the effect of live *LA1* on OA progression	*LA1*	(1) OA; (2) OA + *LA1* 24 days	Histology, Mankin and OARSI scores	DNA sequencing (targeting V3–V4 region of 16s rRNA gene)	*LA1* and butyrate administration ↓pain, cartilage damage, intestinal damage, IL1β, MCP1, TNFα and ↑OCLN, ZO1, *Bifidobacterium*, *Faecalibacterium prausnizii*	*LA1* or butyrate ameliorates OA progression by modulating the gut environment	Cho et al. 2022 [37], USA	High
Probiotics	Balb/c mice, 8 weeks old, NR, NR	HFD	To analyze the correlation between OA and gut microbiota and to evaluate the treatment with *L. paracasei subsp. paracasei M5* (*M5*) and CS for OA prevention	*M5*, CS and CS-*M5*	(1) Control diet + oral gavage saline; (2) HFD + oral gavage saline; (3) HFD + CS; (4) HFD + *M5*; (5) HFD + CS-*M5* 12 weeks	Histology, Mankin score	DNA sequencing (targeting V3–V4 region of 16s rRNA gene)	↑COLII, adiponectin and ↓leptin in HFD + *M5* and HFD + CS-*M5* groups vs. all the other groups	*M5* regulates adipokines levels and gut microbiota composition and inhibits OA development	Song et al. 2020 [38], China	High
ABT	C57BL/6N mice, 8 weeks, male and female, 54	DMM	To evaluate the relationship between gut dysbiosis induced by ABT administration and structural changes in early-stage OA	Ampicillin, neomycin	(1) Control-female; (2) OA-female; (3) ABT-OA-female; (4) Control-male; (5) OA-male; (6) ABT-OA-male 8 weeks	DXA, Micro-CT, histology, IHC	Universal primers and specific bacterial primers, including those for *α-proteobacteria*, *γ-proteobacteria*, *Bacteroidetes*, *Firmicutes* and *Actinobacteria*	ABT-induced intestinal microbiota dysbiosis ↓LPS serum level, TNFα and IL6, which lead to ↓MMP13	Gut microbiome dysbiosis alleviates OA progression	Guan et al. 2020 [27], China	High
ABT	C57BL/6J and TLR5KO mice, NR, male, 10–11 for group	Load-induced OA (cyclic compressive loading)	To evaluate the impact of obesity, metabolic syndrome and gut microbiome on load-induced OA	Ampicillin, neomycin	(1) Control C57BL/6J; (2) TLR5KO + metabolic syndrome; (3) TLR5KO + ABT; (4) C57BL/6J + HFD 4 weeks	Histology, OARSI score	DNA sequencing (targeting 16s rRNA gene)	↓Thickness of subchondral bone plate in TLR5KO + ABT group; ↑LPS, KC, IL10, TNFα and ↓IL6 in C57BL/6J + HFD group vs. TLR5KO + ABT group	Changes in gut microbiota influence the severity of OA	Guss et al. 2019 [39], USA	High
ABT	SD rats, 8 weeks old, male, 32	ABT pretreatment combined with a Try-rich diet (or not)	To evaluate the role of aryl hydrocarbon R in OA and its association with intestinal microbiome	ABT and Try intervention	(1) Sham surgery; (2) OA control; (3) ABT; (4) ABT + Try 9 weeks	Histology, OARSI score	DNA sequencing (targeting V3–V4 region of 16s rRNA gene)	↑Col2A1, SOX9 and ↓MMP13, LPS in ABT group vs. OA control group; ↑AhR, CyP1A1, Col2A1, MMP13 in ABT + Try group vs. OA control and ABT groups	Try supplementation activated intestinal microbiome-related Try metabolism, antagonizing the effects of ABT, exacerbating OA	Chen et al. 2023 [40], China	Low
Non-pharmacological treatments	Wistar rats, 8 weeks old, male, 37	MIA	To evaluate the effect and dose-effect of MS on OA	MS	(1) Healthy control; (2) OA; (3) OA + MS for 2 weeks; (4) OA + MS for 4 weeks; (5) OA + MS for 6 weeks 2, 4, 6 weeks	Histology, Mankin score	DNA sequencing (targeting V3–V4 region of 16s rRNA gene)	↑Chondrocytes, IL10 and ↓IL1β, TNFα in OA + MS for 4 and 6 weeks groups vs. OA and OA + MS for 2 weeks groups	MS treatment regulates intestinal microbiome composition and inflammation, influencing OA progression	Jia et al. 2022 [41], China	Moderate
Non-pharmacological treatments	SD rats, 8 weeks old, male, 30	HFD	To evaluate the effect of acupunctures on gut microbiome and OA recovery	ST36, GB34, and ST36 + GB34 acupuncture treatments	(1) Control chow diet; (2) HFD; (3) HFD + ST36; (4) HFD + GB34; (5) HFD + ST36 + GB34 14 weeks	Histology, Mankin score	DNA sequencing (targeting V3–V4 region of 16s rRNA gene)	↓Cartilage loss, MMP1, MMP13 in HFD + ST36, HFD + GB34 and HFD + ST36 + GB34 groups vs. HDF group; ↓inflammatory cytokines, VEGF, MIP1α, MIP2, MCP1, LPS level, 65 and P-p65 in HFD + ST36 + GB34 group vs. HFD, HFD + ST36 groups	ST36 and GB34 treatments inhibit OA degradation by regulating the lipid metabolism and gut microbiota	Xie et al. 2020 [42], China	High
Non-pharmacological treatments	Wistar rats, 12 weeks, male, 36	MIA	To analyze the effect of MS on the intestinal flora during OA	MS or DS	(1) Healthy control; (2) OA; (3) OA + MS; (4) OA + DS 4 weeks	Histology	DNA sequencing (targeting V3–V4 region of 16s rRNA gene)	↓IL1β, TNFα, LPS in MS and DS groups	MS reduces OA cartilage damage by regulating the composition of intestinal flora	Chen et al. 2020 [43], China	Low
HFD	Rats, 8–12 weeks old, NR, 32	HFD	To evaluate the relationship between inflammation, gut microbiota, and metabolic OA	No treatment	(1) Diet-induced obesity; (2) Chow diet 28 weeks	Mankin score	DNA sequencing (targeting 16s rRNA gene)	↑Mankin score, synovial fluid and serum analytes, LPS in diet-induced obesity group vs. chow diet group	The high presence of *Lactobacillus species* (spp.) and *Methanobrevibacter* spp. had a strong predictive relationship with OA Mankin score	Collins et al. 2015 [44], Canada	High
HFD	C57BL/6J mice, 12 weeks old, male, 54	HFD	To evaluate if wheel-running exercise prevents OA induced by HFD by reducing LPS from intestinal microorganisms	No treatments	(1) Control; (2) HFD 8 weeks	Histology, Mankin score, histochemistry	DNA sequencing (targeting 16s rRNA gene)	HFD treatment ↓gut microbial diversity, gut barrier-protecting bacteria and ↑endotoxin-producing bacteria. The voluntary wheel running ↓TLR4, MMP13 and LPS levels in blood and synovial fluid	Exercising can remodel gut microbial ecosystems, reduce the circulating levels of LPS, contributing to the relief of chronic inflammation and OA	Li et al. 2021 [45], China	Low
Anti-hypertensive	SD rats, 6 weeks old, NR, 24	DOCA, OA and hypertensive	Impact of hypertension on the articular cartilage and subchondral bone and the therapeutical effect of senescence removal by anti-hypertensive drug captopril	Captopril	(1) Healthy control; (2) DOCA; (3) DOCA + captopril 14 weeks	Micro-CT, histology, IHC	DNA sequencing (targeting V3–V4 region of 16s rRNA gene)	↑p16 and proteoglycan loss in DOCA group vs. control; ↓senescence cells in DOCA + captopril group	Captopril has an anti-senolytic effect and decreases cartilage degeneration by restoring gut microbial structure	Chan et al. 2022 [46], China	High
No treatments	Rhesus macaque, 6–15 years old, female, 20	Spontaneous OA	To explore the relationship between OA and intestinal microbiota	No treatment	(1) Healthy control; (2) OA NR	MRI	DNA extraction from fecal samples and metagenomics analysis	↑*LA* in OA monkeys; ↑*Prevotella* and *Ruminococcus* in non-OA monkeys	The diversity and composition of intestinal microbiota in monkeys with OA are different compared to the normal monkeys	Yan et al. 2021 [47], China	High
No treatments	Mice LD and littermate control (WT), 16 weeks old, male and female, 5–10 for sex and group	DMM	To define a relationship between knee cartilage damage and gut microbiota	No treatment	LD and WT mice, fed HFD/chow or rescued with fat implantation 13 weeks	Histology, Mankin score	DNA sequencing (targeting 16s rRNA gene)	↑Synovial fluid LPS levels in HFD WT mice vs. all groups; ↓*Bacteroidetes*:*Firmicutes* ratio of the gut microbiota in HFD and OA-rescued animals vs. chow	Causal relationships between gut microbiome and cartilage health, independent of diet or adiposity	Collins et al. 2021 [48], USA	High
No treatments	C57BL/6J and TLR5KO mice, 4 weeks old, male, 88	Load-induced OA	To evaluate the influence of obesity on cartilage degeneration	No treatment	(1) Severe obesity (C57Bl6/J + HFD) (2) mild obesity (TLR5KO + standard chow) (3) normal adiposity (C57Bl6/J + standard chow) (4) TLR5KO + ABT + normal diet 6 weeks	Micro-CT, histology	DNA sequencing (targeting 16s rRNA gene)	No differences in the severity of cartilage degeneration among groups, constituents of the gut microbiota (*Verrucomicrobia*, *Proteobacteria*, *Tenericutes* and *Actinobacteria*) differed among groups	No association between components of the gut microbiota with OA	Luna et al. 2021 [49], USA	High
No treatments	C57BL/6J mice, NR, male, 43	DMM	To evaluate the contribution of gut microbiota to develop OA after joint injury	No treatment	(1) C57BL/6J GF; (2) C57BL/6J SPF (two groups: 13.5 weeks age at DMM and 43 weeks age at DMM) 8 weeks	Histology, OARSI score	DNA sequencing (targeting V3–V4 region of 16s rRNA gene)	↓Cartilage damage and proteoglycan loss in GF group vs. SPF groups	Gut microbiota promote OA progression	Ulici et al. 2018 [50], USA	High
No treatments	C57BL/6 mice GF and conventional, 20 weeks old, NR, 50	ACL injury	To evaluate the gut microbiota effect on OA progression	No treatment	(1) GF OA; (2) GF contralateral; (3) GF naïve; (4) conventional OA; (5) conventional contralateral; (6) conventional naïve 1 week	Micro-CT	Global metabolomic profiling	↑Trabecular bone volume and ↓trabecular bone loss, sensitive to injury in GF groups vs. conventional groups	Gut microbiota promotes OA development	Hahn et al. 2021 [51], USA	High
No treatments	Hartley guinea pigs, 7 weeks old, male, 36	Spontaneous OA	To evaluate sedentary lifestyle contribution to OA incidence and severity	No treatment	(1) Sedentary; (2) physically active 22 weeks	Histology	DNA sequencing (targeting V4 region of 16s rRNA gene)	No taxa distinguished the microbial composition of samples collected from sedentary and physically active animals	Gut microbial communities of sedentary and physically active OA animals were indistinguishable	Wallace et al. 2019 [52], USA	Moderate

Abbreviations: ABT = antibiotic; ACL = anterior cruciate ligament; CS = chondroitin sulfate; COL2A1 = collagen type II alpha 1 chain; COLII = collagen type II; DMM = destabilization of the medial meniscus; DOCA = deoxy-corticosterone acetate; DS = diclofenac sodium; DXA = dual-energy X-ray absorptiometry; ELISA = enzyme-linked immunosorbent assay; GF = germ free; GFRα3 = GDNF family receptor alpha-3; HFD = high-fat diet; HFS = high-fat high-sucrose; IHC = immunohistochemistry; IL = interleukin; LA1 = lactobacillus acidophilus; LD = lipodystrophic; LPS = lipopolysaccharide; mEVs = milk-derived extracellular vesicles; MIA = monosodium iodoacetate; Micro-CT = micro-computed tomography; MIP = macrophage inflammatory proteins; MCP1 = monocyte chemoattractant protein-1; MMP = matrix metallopeptidase; MS = moxibustion; MY = Miya; NFKB = nuclear factor kappa B; NGF = nerve growth factor; NLRP3 = NLR family pyrin domain containing 3 protein; NR = not reported; OA = osteoarthritis; OARSI = Osteoarthritis Research Society International; OCLN = occluding; PBS = phosphate-buffered saline; PGE2 = prostaglandin E2; PMM = partial meniscectomy; R = receptor; RUNX2 = RUNX family transcription factor 2; SCFAs = short-chain fatty acid; SD = Sprague–Dawley; SPF = specific pathogen free; TLR4 = Toll-like receptor 4; TNFα = tumor necrosis factor alpha; Try = tryptophan; VEGF = Vascular Endothelial Growth Factor; ZO1 = Zonula occludens-1; ↑ = increase; ↓ = decrease.

**Table 3 ijms-25-00143-t003:** Main characteristics of clinical studies included in the review.

Treatment Classification	Study Design	Age/Gender/Number	OA Assessment Methods	Study Aim	Treatment	Groups	Follow-Up or Experimental Time	Microbiota Assessment	Main Results	Ref., Year, Country	ROBINS I Risk of Bias
Probiotics	RCT	Mean age: 50 ± 90 yrs, NR, 60	KL	Effect of probiotics on OA pain	Probiotic formulation of *LA casei*	(1) Healthy control; (2) OA	6 weeks	VAS, pressure pain threshold, IL6, TNFα, IL6 R, IL1 R, CRP, DNA sequencing (targeting 16s rRNA gene)	Probiotics and conservative treatment improve OA-related pain	Pedersini et al. 2021 [53], Italy	Low
Probiotics	Trial of individual effects	67 yrs, female, 1	NR	Effectiveness of probiotics on OA pain	*LA rhamnosus*, *Saccharomyces cerevisiae* (*boulardii*) and *Bifidobacterium animalis ssp lactis* vs. placebo treatment	(1) OA	32 weeks	VAS, GHQ-12, PSFS, CDSA	Probiotic use reduces OA pain	Taye et al. 2020 [54], Australia	Low
Non-pharmacological treatments	Multicentric RCT	45–75 yrs, NR, 90	X-ray	Effect of EA on gut microbiota in OA pts	EA	(1) Healthy controls; (2) OA sham acupuncture; (3) OA EA	8 weeks	DNA sequencing (targeting 16s rRNA gene)	↑*Streptococcus* significant associated with OA severity	Wang et al. 2021 [55], China	Low
No treatments	Retrospective	NR, NR, 18345	NR	Link between gut microbiota imbalance and OA progression	No treatments	(1) OA	NR	DNA sequencing (targeting 16s rRNA gene)	*Methanobacteriaceae* and *desulfoovibrionales* onegatively correlated with knee OA risk	Yu et al. 2021 [56], China	Low
No treatments	Prospective	56.9 yrs, male (606), female (821), 1427	KL, X-ray, WOMAC	Relationship between joint pain and microbiome composition in OA	No treatments	(1) Healthy control; (2) OA	NR	Analysis of taxonomic profiling of gastrointestinal microbiota from stool samples	↑*Streptococcus* spp. significantly associated with ↑WOMAC in OA pts	Boer et al. 2019 [57], Netherlands	Low
No treatments	Prospective	NR, female (34) and male (10) OA pts, female (28) and male (18) control pts	NR	Characterizations of the gut bacteriome, mycobiome, and virome in OA pts	No treatments	(1) Healthy control; (2) OA	NR	DNA extraction and whole-metagenome shotgun sequencing	Gut microbiome of OA pts completely altered compared to that of healthy individuals	Chen et al. 2021 [58], China	Low
No treatments	Observational	Mean age: 45.5 ± 10.2 yrs, NR, 24	KL, WOMAC	Gut microbiome composition in OA pts and normal individuals with or without VDD	No treatments	(1) OA-VDD (*n* = 7); (2) OA (*n* = 4); (3) VDD (*n* = 7); (4) NVD (*n* = 6)	6 months	DNA sequencing (targeting V3-V4 region of 16s rRNA gene)	Association between gut microbiome, vitamin D and knee OA	Ramasamy et al. 2021 [59], India	Low
No treatments	Cross-sectional	Mean age: 53.69 ± 4.58 yrs, NR, 78	KL	Gut microbiota and metabolites effect in OA	No treatments	(1) Healthy control; (2) OA	NR	Albumin, hemoglobin, pro-albumin, superoxide dismutase, monoamine oxidase, glutathione reductase, total iron binding serum concentration analyses	Protective effect of gut microbiota and its metabolite capsiate on ferroptosis-relative OA	Guan et al. 2023 [60], China	Low
No treatments	Retrospective	>45 yrs, NR, 92	KL	Perturbations in gut microbial composition and the gut metabolome linked to individuals with obesity and OA	No treatments	(1) Healthy control; (2) OA	NR	Untargeted fecal metabolomics analysis	Perturbations in leukotriene metabolism, and changes in microbial metabolites modify OA proteolysis	Rushing et al. 2022 [61], USA	Low
No treatments	Retrospective	Mean age: 65.0 ± 7.7 yrs, NR, 57	NR	Relationship between gut microbiome and OA in the older female adults	No treatments	(1) Healthy control; (2) OA	NR	DNA sequencing (targeting 16s rRNA gene)	Alterations in the gut microbial composition and function in OA	Chen et al. 2021 [62], China	Low
No treatments	Prospective	<75 yrs, NR, 182	NR	Gut microbiome and risk for OA	No treatments	(1) Healthy control; (2) OA	NR	DNA sequencing (targeting 16s rRNA gene)	Diversity and richness of the gut microbiome ↓in overweight OA pts	Wang et al. 2021 [63], China	Low
No treatments	Cross-sectional	>55 yrs, NR, 70	KL	Correlation among plasma microbiota, LPS, and obesity-associated OA	NR	(1) OA	NR	LPS level and DNA sequencing (targeting 16s rRNA gene)	Correlation between serum LPS and plasma microbiome	Arbeeva et al. 2022 [64], USA	Low
No treatments	Prospective	>45 yrs, NR, 92	KL	Role of dysbiosis in obesity-associated OA	No treatments	(1) Healthy control; (2) OA	NR	DNA sequencing (targeting 16s rRNA gene), cytokine and LPS measures	↑Intestinal permeability contribute to OA development associated with obesity	Loeser et al. 2022 [65], USA	Low
No treatments	Two-sample Mendelian randomization	NR, NR, 11400	KL	Correlation between intestinal microbiota and OA occurrence	No treatments	(1) Healthy control; (2) OA	NR	NR	No association between intestinal microbiome and OA	Lee et al. 2021 [66], Korea	Low

Abbreviations: CDSA = Comprehensive Digestive Stool Analysis; CRP = C-reactive protein; EA = electroacupuncture; GHQ = General Health Questionnaire; KL = Kellgren–Lawrence; IL = interleukin; LA = Lactobacillus; LPS = lipopolysaccharide; NR = not reported; NVD = normal vitamin D; OA = osteoarthritis; PSFS = Patient-Specific Functional Scale; pts = patients; R = receptor; RCT = randomized controlled trial; TNFα = tumor necrosis factor alpha; VAS = Visual Analogue Scale; VDD = vitamin D deficiency; WOMAC = Western Ontario McMaster OsteoArthritis Index; yrs = years.

## Data Availability

Not applicable.

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
