# Peer review of "The Complex Interplay between the Gut Microbiome and Osteoarthritis: A Systematic Review on Potential Correlations and Therapeutic Approaches"

_ijms, 2023, doi:10.3390/ijms25010143_

Round 1
Reviewer 1 Report
Comments and Suggestions for Authors
1. This article offers a comprehensive overview of preclinical and clinical studies exploring the potential therapeutic applications of probiotics and prebiotics. While the attached article addresses the main topic, there appears to be insufficient processing of genetic background in the work.
2. Consistency in the first use of abbreviations is crucial in the abstract. For instance, consistently use "OA" and provide the initial definition of "HFD."
3. In the introduction, the authors should incorporate information on MMPs and type 2 collagen. Additionally, consider introducing the role of VEGF and its impact on OA through probiotics and prebiotics.
4. Figure 1 requires improvement for better clarity and understanding.
5. Maintain consistency, such as using abbreviations in line 166 and 190. Ensure uniform use of abbreviations in lines 198-200.
6. In Table 1, include the names of short-chain fatty acids.
7. Tables need improvement for a more polished appearance.
8. Explore further opportunities to enhance the presentation of the link between the VEGFR-1 gene and probiotics. Consider creating a table summarizing the effects of probiotics, prebiotics, or symbiotics on the expression of the VEGFR-1 gene and the associated outcomes of altered gene expression.
9. In the paragraph discussing the "Role of Probiotics in Inflammatory Markers in OA," consider creating a table to elucidate and improve the explanation of the gene-probiotic-OA relationship.
10. Overall, the work is intriguing and has the potential to generate new research ideas. Incorporate the above suggestions to enhance the understanding of the gastrointestinal microbiota's impact on OA pathogenesis.
11. Grammar improvement is necessary throughout the manuscript.

Need to check the grammar
Author Response
- This article offers a comprehensive overview of preclinical and clinical studies exploring the potential therapeutic applications of probiotics and prebiotics. While the attached article addresses the main topic, there appears to be insufficient processing of genetic background in the work.
We thank the reviewer for the valuable suggestion. As recommended, we included the association between microbiota and genetic background in the introduction section.
- Consistency in the first use of abbreviations is crucial in the abstract. For instance, consistently use "OA" and provide the initial definition of "HFD."
We thank the reviewer for the suggestion. We standardized and corrected all abbreviations in the text.
- In the introduction, the authors should incorporate information on MMPs and type 2 collagen. Additionally, consider introducing the role of VEGF and its impact on OA through probiotics and prebiotics.
As suggested by the reviewer we added in the introduction section the link between VEGF and OA.
- Figure 1 requires improvement for better clarity and understanding.
As suggested by the reviewer, we modified Figure 1 for better clarity and understanding.
- Maintain consistency, such as using abbreviations in line 166 and 190. Ensure uniform use of abbreviations in lines 198-200.
We standardized and corrected all abbreviations in the text.
- In Table 1, include the names of short-chain fatty acids.
As suggested by the reviewer, we have defined all abbreviations in the manuscript and added an abbreviations legend after each table.
- Tables need improvement for a more polished appearance.
As suggested, we tried to improve the tables.
- Explore further opportunities to enhance the presentation of the link between the VEGFR-1 gene and probiotics. Consider creating a table summarizing the effects of probiotics, prebiotics, or symbiotics on the expression of the VEGFR-1 gene and the associated outcomes of altered gene expression.
As indicated by the reviewer, we have delved deeper and added additional information in the results section on the link between the VEGFR gene and probiotics. However, due to the limited number of studies addressing this novel and impactful aspect, adding an additional table proves challenging. Furthermore, the current tables still depict the role of VEGF at the local level in the presence of various treatments.
- In the paragraph discussing the "Role of Probiotics in Inflammatory Markers in OA," consider creating a table to elucidate and improve the explanation of the gene-probiotic-OA relationship.
We added and discussed the role of probiotics in inflammatory markers in OA in the discussion section, considering that experimental data from preclinical studies and emerging trends from clinical studies suggest that probiotics may benefit patients with OA pain through positive gut microbiota modulation and attenuating low-grade inflammation via multiple pathways. Further advancement of knowledge in this area will undoubtedly pave the way for the development of probiotic strains that can be used to improve treatment outcomes in OA and reduce the huge impact of this serious disease on healthcare systems worldwide.
- Overall, the work is intriguing and has the potential to generate new research ideas. Incorporate the above suggestions to enhance the understanding of the gastrointestinal microbiota's impact on OA pathogenesis.
We thank the reviewer and as suggested, we modified the manuscript adding more information on genetic background and on the role of VEGF and its impact on OA through probiotics and prebiotics.
- Grammar improvement is necessary throughout the manuscript.
We have extensively revised the manuscript from a grammatical point of view.
Reviewer 2 Report
Comments and Suggestions for Authors Manuscript ID: IJMS-2733485
Dear Authors,
A nice attempt on analyzing the interplay between OA and gut micribiota. Here are my queries and suggestions:
Use italics at appropriate places
Sub-headings should be checked
Lot of Formatting errors, please correct them
Clarify "the effect of a high -fat diet on OA"
Lines : 105 - 108
Fig 1: Looked good, but there was no insightful information about the realtion between OA and Gut microbiota. OA joint? or Fibula? check this term. Place the labels at corresponding locations.
My suggestion would be you can either try to bring in the "relation" in the display or skip the figure.
Lines 242 - 244: Check the sentence
Some specific bacteria were associated with 333 lower risk of OA, add the species name
Line 298: review?
Line 299: mention them "other bacterial species"
Where is the conclusion section, please add
Figs 3 and 4: Increase the clarity of legends and labels
Lines 257 - 258: rewrite the sentence
Section 2.3.1 lacks coherence, modify that, convey the results clearly
Lines 164 - 165: was it acupuncture or electroacupuncture?
Highlight the overall outcome of your review, explicitly.
Comments on the Quality of English LanguageMust improve the sentences and writing style, also correct the errors. Authors should focus on the mechanisms, especially.
Author Response
Dear Authors,
A nice attempt on analyzing the interplay between OA and gut micribiota. Here are my queries and suggestions:
Use italics at appropriate places
We corrected.
Sub-headings should be checked
We checked and corrected.
Lot of Formatting errors, please correct them
We thank the reviewer for the suggestion, and we extensively revised the manuscript from a grammatical point of view.
Clarify "the effect of a high -fat diet on OA"
As suggested by the reviewer, we clarified the effect of a high-fat diet on OA.
Lines : 105 - 108
Fig 1: Looked good, but there was no insightful information about the realtion between OA and Gut microbiota. OA joint? or Fibula? check this term. Place the labels at corresponding locations.
My suggestion would be you can either try to bring in the "relation" in the display or skip the figure.
As suggested by the reviewer, we modified Figure 1 for better clarity and understanding.
Lines 242 - 244: Check the sentence
We modified the sentence as requested.
Some specific bacteria were associated with 333 lower risk of OA, add the species name
We specified the bacteria associated to a lower OA risk.
Line 298: review?
We thank the reviewer, and we corrected the error.
Line 299: mention them "other bacterial species"
We added the other bacterial species involved in OA.
Where is the conclusion section, please add
As suggested by the reviewer we added the conclusion section.
Figs 3 and 4: Increase the clarity of legends and labels
As suggested, we clarified the legend and added further information in the Supplementary Materials section.
Lines 257 - 258: rewrite the sentence
We corrected the sentence.
Section 2.3.1 lacks coherence, modify that, convey the results clearly
As suggested, we modified the section 2.3.1.
Lines 164 - 165: was it acupuncture or electroacupuncture?
It was electroacupuncture, we corrected in the text and table.
Highlight the overall outcome of your review, explicitly.
We explicated the overall outcome in the conclusion section that we added after the discussion section.
Comments on the Quality of English Language
Must improve the sentences and writing style, also correct the errors. Authors should focus on the mechanisms, especially.
We have extensively revised the manuscript from a grammatical point of view.
Reviewer 3 Report
Comments and Suggestions for Authors
Marchese et al. systematically reviewed the effects of therapy in targeting OA via microbiome regulation. Please refer to my comments:
1) One of the main concerns is the single author in screening (line 427) and extracting the data/records (line 437). According to Cochrane Review guidance (https://www.cochranelibrary.com/cdsr/about-cdsr; DOI:10.1016/j.jclinepi.2017.08.002; 10.21037/atm-22-6305) the article searching, screening and extraction cannot be done by single author. Or else it is not entitled to be named as a systematic review. This is also because one cannot avoid the high risk of bias, and error in searching and reporting if it is done by a single author. Besides, this is not coherent with the author contributions (lines 459-462). Thank you for the truthful writing and kindly repeat the searching, screening and extraction with at least 2 independent authors. Update the necessary info and submit it again.
2) Kindly follow the journal format, for example the abstract and heading/subheading sequences.
3) Be consistent with the abbreviation, use them if necessary, define and use them consistently throughout the manuscript. Please check EV, DOCA, OA, HFD, etc.
4) Title: the title does not directly reflect the work done. The authors also emphasized the effects of treatment/intervention, not merely microbiota
5) Introduction: Try to be more concise with the topic. Simply the inflammation and other risk factors. Should explain more about the relationship between gut microbiota and OA. Besides, need to mention the ways that can regulate or affect the microbiota population
6) Introduction: Research gap is not clearly defined. I understand that the authors particularly interested in anti-OA therapy via modulating microbiota. But I will suggest the authors to emphasize the involvement of microbiota and its modulation as part of the known anti-OA drugs. Some of the agents like quercetin, EVs, prebiotics, bacteria and other supplements do not have the primary effects in anti-OA. Besides, we are not sure if the therapeutic effects are mainly through the microbiota or other mechanisms. Again based on the results and discussion, the authors emphasized on the microbiota changes and related them with anti-OA outcomes. What makes it different from the previous reviews then (line 80-84)?
7) The species name should be italicised.
8) Figure 2, what does it mean by "excluded by title and abstract" ? The screening should be done after remove the duplicate, not before.
9) Line 396: Wrong citation for PRISMA guidelines and checklist. Why not using PRISMA 2020 guidelines as there are several relevant reviews?
10) What are the inclusion and exclusion criteria? Need to have a clear cut, especially on the intervention or specific therapies (line 83).
11) Table 1: Please proofread as some information is wrong, for example, use DNA sequencing for 16s-RNA.
12) Table 1: Please include the respective critical appraisal score for each article. How the authors identify or judge the quality of an article is high or poor (line 357)?
13) Table 3: Duplicate entry for PubMed. Why limit the search to 10 years? (line 386-387)
14) Result (line 150-153), please cite the articles, not their number.
Comments on the Quality of English LanguageNo issue
Author Response
Comments and Suggestions for Authors
Marchese et al. systematically reviewed the effects of therapy in targeting OA via microbiome regulation. Please refer to my comments:
1) One of the main concerns is the single author in screening (line 427) and extracting the data/records (line 437). According to Cochrane Review guidance (https://www.cochranelibrary.com/cdsr/about-cdsr; DOI:10.1016/j.jclinepi.2017.08.002; 10.21037/atm-22-6305) the article searching, screening and extraction cannot be done by single author. Or else it is not entitled to be named as a systematic review. This is also because one cannot avoid the high risk of bias, and error in searching and reporting if it is done by a single author. Besides, this is not coherent with the author contributions (lines 459-462). Thank you for the truthful writing and kindly repeat the searching, screening and extraction with at least 2 independent authors. Update the necessary info and submit it again.
We thank the reviewer for the valuable suggestion. As recommended, we have included a second author (FS) in the systematic evaluation. This had already reviewed all the works and collaborated with the first author in making decisions during the manuscript preparation and writing. Thus, as a result, we have modified the manuscript as requested.
2) Kindly follow the journal format, for example the abstract and heading/subheading sequences.
We modified the manuscript following the journal format (e.g. abstract and heading/subheading sequences):
3) Be consistent with the abbreviation, use them if necessary, define and use them consistently throughout the manuscript. Please check EV, DOCA, OA, HFD, etc.
As suggested by the reviewer, we have defined all abbreviations in the manuscript and added an abbreviations legend after each table.
4) Title: the title does not directly reflect the work done. The authors also emphasized the effects of treatment/intervention, not merely microbiota
After the reviewer suggestion we modified the manuscript title: “The complex interplay between the gut microbiome and osteoarthritis: a systematic review on potential correlations and therapeutic approaches”.
5) Introduction: Try to be more concise with the topic. Simply the inflammation and other risk factors. Should explain more about the relationship between gut microbiota and OA. Besides, need to mention the ways that can regulate or affect the microbiota population.
As requested by the reviewer we modified the introduction section trying to be more concise with the topic, to explain more about the relationship between gut microbiota and OA and to mention the ways that can regulate or affect the microbiota population.
6) Introduction: Research gap is not clearly defined. I understand that the authors particularly interested in anti-OA therapy via modulating microbiota. But I will suggest the authors to emphasize the involvement of microbiota and its modulation as part of the known anti-OA drugs. Some of the agents like quercetin, EVs, prebiotics, bacteria and other supplements do not have the primary effects in anti-OA. Besides, we are not sure if the therapeutic effects are mainly through the microbiota or other mechanisms. Again based on the results and discussion, the authors emphasized on the microbiota changes and related them with anti-OA outcomes. What makes it different from the previous reviews then (line 80-84)?
As suggested by the reviewer we clarified the research gap in the introduction section, emphasizing the involvement of microbiota and its modulation as part of anti-OA treatments. It is precisely this aspect that sets our reviews apart from those previously undertaken. In fact, previous reviews analyzed the crosstalk between OA and the intestinal microbiota, primarily focusing on gut microbiome composition, OA severity and pain, inflammatory factors, and intestinal permeability, but none of them explored microbiota and its modulation as part of anti-OA treatments.
7) The species name should be italicised.
We italicized the species name.
8) Figure 2, what does it mean by "excluded by title and abstract"? The screening should be done after remove the duplicate, not before.
We corrected Figure 2.
9) Line 396: Wrong citation for PRISMA guidelines and checklist. Why not using PRISMA 2020 guidelines as there are several relevant reviews?
We corrected the reference and added the PRISMA 2020 guidelines.
10) What are the inclusion and exclusion criteria? Need to have a clear cut, especially on the intervention or specific therapies (line 83).
As suggested, we tried to better specified inclusion and exclusion criteria. However, for intervention or specific therapies we eliminated only articles that did not specified intervention or specific therapies used and articles in which data were not accessible or missing.
11) Table 1: Please proofread as some information is wrong, for example, use DNA sequencing for 16s-RNA.
We proofread table 1 as suggested.
12) Table 1: Please include the respective critical appraisal score for each article. How the authors identify or judge the quality of an article is high or poor (line 357)?
We include the critical appraisal score for each article (preclinical and clinical) in the Supplementary Materials. The methodological quality of the included clinical studies was assessed following the ROBINS-I tool while the methodological quality of the included in vivo studies was done according to the SYRCLE tool.
13) Table 3: Duplicate entry for PubMed. Why limit the search to 10 years? (line 386-387)
The decision to limit this systematic review to the last 10 years was influenced by several factors. Firstly, it allowed for the incorporation of up-to-date and relevant evidence in light of the significant changes that often occur in scientific knowledge and clinical practices over a decade. Additionally, more recent studies are considered more clinically relevant as they may reflect advances in therapies, diagnostic technologies, or treatment paradigms. Finally, the choice to focus on the last 10 years ensured that the included studies are in line with current research methodologies and adhere to contemporary publication standards. Furthermore, limiting the time frame has certainly helped in the synthesis of a more current body of evidence, also simplifying the systematic review process.
14) Result (line 150-153), please cite the articles, not their number.
As suggested by the reviewer, we cited the articles in the result section.
Reviewer 4 Report
Comments and Suggestions for Authors
The paper is a systematic review of the current literature on the possible correlation between gut microbiota and osteoarthritis, specifically examining the feasibility of microbiota-targeted therapies for the treatment of osteoarthritis. It is reported that 29 studies (24 preclinical and 15 clinical.... the sum is 39!!) were identified and used to synthesise the results. The preclinical studies were mainly focused on modifying the microenvironment of the gut microbiota, using dietary supplements, probiotics and prebiotics to modify osteoarthritis status.
I have some major concerns for the author that I will split according to the section:
INTRODUCTION
Please make sure that the reader already knows what the gut-joint axis is by the introduction. It is not mentioned at all.
Line 74. Treatments like platelet-rich plasma belong to the group of the DMOADs. Please add and clarify this concept.
Line 89. Add reference
Figure 1. The figure doesn’t make any better the explanation of the axis gut- joint and appears to me redundant. While I like the graphical concept, I would suggest removing unnecessary information (is it really needed to put on the picture the 6 phyla?) and adding more important details regarding what shapes this axis, such as the cytokines involved in inflammation in the context of dysbiosis.
METHODS
I would advise to move the methods section before the discussion. It is very confusing.
What is the Prospero registration number?
Please remove the description of the Boolean keywords used from the flowchart and move it/leave it to the main text. Also, what you write here is different from the keywords you describe in the text ( you add microbiome and more)
From my search, I get on Pubmed more than 124 papers when I use microbio*. Hence some papers may have been left behind. I suggest revising the keywords.
TABLES.
I find them hard to read.
DISCUSSION
Please rearrange in preclinical followed by clinical. Right now, instead, I see a specific paragraph about lactobacillus then pre clinical and clinical data are mixed up.
Line 298. “In this revie” please remove and change to “ In this paper”
Line 306. Reference 70 is missing!!!
Line 333. “Furthermore some specific bacteria….” What bacteria? Please be specific and also add references throughout the entire discussion.
Comments on the Quality of English Language
The quality of English is good.
Author Response
The paper is a systematic review of the current literature on the possible correlation between gut microbiota and osteoarthritis, specifically examining the feasibility of microbiota-targeted therapies for the treatment of osteoarthritis. It is reported that 29 studies (24 preclinical and 15 clinical.... the sum is 39!!) were identified and used to synthesise the results. The preclinical studies were mainly focused on modifying the microenvironment of the gut microbiota, using dietary supplements, probiotics and prebiotics to modify osteoarthritis status.
We thank the reviewer for the valuable suggestion. We correct the number of studies.
I have some major concerns for the author that I will split according to the section:
INTRODUCTION
Please make sure that the reader already knows what the gut-joint axis is by the introduction. It is not mentioned at all.
As suggested by the reviewer we added the concept of gut-joint axis in the introduction section.
Line 74. Treatments like platelet-rich plasma belong to the group of the DMOADs. Please add and clarify this concept.
As suggested by the reviewer we added and clarified this concept in the introduction section.
Line 89. Add reference
We added the reference.
Figure 1. The figure doesn’t make any better the explanation of the axis gut- joint and appears to me redundant. While I like the graphical concept, I would suggest removing unnecessary information (is it really needed to put on the picture the 6 phyla?) and adding more important details regarding what shapes this axis, such as the cytokines involved in inflammation in the context of dysbiosis.
As suggested by the reviewer, we modified Figure 1 for better clarity and understanding.
METHODS
I would advise to move the methods section before the discussion. It is very confusing.
As suggested, we moved the methods section before the discussion.
What is the Prospero registration number?
Before starting this review and to avoid overlap with other ongoing review studies, we searched PROSPERO for any similar reviews. However, we have not registered the review on PROSPERO, we will do it.
Please remove the description of the Boolean keywords used from the flowchart and move it/leave it to the main text. Also, what you write here is different from the keywords you describe in the text ( you add microbiome and more)
We changed the Boolean keywords. However, we have included two that were not present in the flowchart (i.e. preclinical and clinical studies; systematic review) to help readers quickly understand what is present in the review.
From my search, I get on Pubmed more than 124 papers when I use microbio*. Hence some papers may have been left behind. I suggest revising the keywords.
We attempted to incorporate different keywords, adhering to the inclusion and exclusion criteria outlined in our review. However, the number of papers remained unchanged. Introducing additional keywords does increase the overall count, but a significant portion of these papers are not pertinent to the objectives of our review.
TABLES.
I find them hard to read.
As suggested, we tried to simplify the tables in the manuscript.
DISCUSSION
Please rearrange in preclinical followed by clinical. Right now, instead, I see a specific paragraph about lactobacillus then pre clinical and clinical data are mixed up.
As suggested by the reviewer we rearranged the discussion section.
Line 298. “In this revie” please remove and change to “ In this paper”
We corrected.
Line 306. Reference 70 is missing!!!
We corrected the references.
Line 333. “Furthermore some specific bacteria….” What bacteria? Please be specific and also add references throughout the entire discussion.
As suggested by the reviewer we modified the discussion section.
Round 2
Reviewer 1 Report
Comments and Suggestions for Authors
Manuscript ready for publication
Author Response
Manuscript ready for publication
We thank the reviewer.
Reviewer 2 Report
Comments and Suggestions for Authors
Dear Authors,
I can see that you have made changes as per my suggestions in your manuscript.
Still, there were so many formatting errors.
Figures 3 and 4: Should increase the clarity.
Revision has improved your work.
Comments on the Quality of English LanguageMinor editing in the language needed.
Author Response
Dear Authors,
I can see that you have made changes as per my suggestions in your manuscript.
Still, there were so many formatting errors.
We corrected the formatting errors.
Figures 3 and 4: Should increase the clarity.
As suggested by the reviewer we increase the resolution of Figure 3 and 4 to 300dpi.
Revision has improved your work.
Comments on the Quality of English Language
Minor editing in the language needed.
We checked again and corrected the manuscript for English Language.
Reviewer 3 Report
Comments and Suggestions for Authors
Follow-up comments:
3) Be consistent with the abbreviation, use them if necessary, define and use them consistently throughout the manuscript. Please check EV, DOCA, OA, HFD, etc.
As suggested by the reviewer, we have defined all abbreviations in the manuscript and added an abbreviations legend after each table.
Follow-up comments: Thank you the authors for the amendment. However it is not consistent. Kindly review and remove those abbreviations that used only once. Besides, make sure define all the abbreviations and used them consistently throughout the text. Please check the Evs.
8) Figure 2, what does it mean by "excluded by title and abstract"? The screening should be done after remove the duplicate, not before.
We corrected Figure 2.
Follow-up comment: Figure 2 is not corrected as stated. Why exclude 142 articles before removing the duplicates?? Please correct it.
9) Line 396: Wrong citation for PRISMA guidelines and checklist. Why not using PRISMA 2020 guidelines as there are several relevant reviews?
We corrected the reference and added the PRISMA 2020 guidelines.
Follow-up comments (line 183): Thank you the authors for the correction. However, PRISMA 2020 is not merely a citation. The authors neeed to compliant with its guidelines and provide the 27-item checklist. Additionally, PRISMA flow diagram (Figure 2) should be updated according to PRISMA 2020. Please double check whether the authors followed the PRISMA 2020 or 2009? Why not following PRISMA 2020? Please attach the PRISMA checklist
10) What are the inclusion and exclusion criteria? Need to have a clear cut, especially on the intervention or specific therapies (line 83).
As suggested, we tried to better specified inclusion and exclusion criteria. However, for intervention or specific therapies we eliminated only articles that did not specified intervention or specific therapies used and articles in which data were not accessible or missing.
Follow-up comments (line 232 and Figure 2): Kindly state the inclusion and exclusion criteria. Incorporate selecting criteria in Figure 2 is not suitable.
11) Table 1: Please proofread as some information is wrong, for example, use DNA sequencing for 16s-RNA.
We proofread table 1 as suggested.
Follow-up comments: Table 2 (after renamed) is still found with several errors. DNA sequencing for 16s rDNA, not RNA. Kindly use the consistent citation/referencing style for tables.
12) Table 1: Please include the respective critical appraisal score for each article. How the authors identify or judge the quality of an article is high or poor (line 357)?
We include the critical appraisal score for each article (preclinical and clinical) in the Supplementary Materials. The methodological quality of the included clinical studies was assessed following the ROBINS-I tool while the methodological quality of the included in vivo studies was done according to the SYRCLE tool.
Follow-up comment: The authors should include the overall scoring for ROBINS and SYRCLE in the Table 2 and 3 (after renamed).
13) Table 3: Duplicate entry for PubMed. Why limit the search to 10 years? (line 386-387)
The decision to limit this systematic review to the last 10 years was influenced by several factors. Firstly, it allowed for the incorporation of up-to-date and relevant evidence in light of the significant changes that often occur in scientific knowledge and clinical practices over a decade. Additionally, more recent studies are considered more clinically relevant as they may reflect advances in therapies, diagnostic technologies, or treatment paradigms. Finally, the choice to focus on the last 10 years ensured that the included studies are in line with current research methodologies and adhere to contemporary publication standards. Furthermore, limiting the time frame has certainly helped in the synthesis of a more current body of evidence, also simplifying the systematic review process.
Follow-up comment: The duplicate entry for PubMed (Table 1) is not corrected.
14) Result (line 150-153), please cite the articles, not their number.
As suggested by the reviewer, we cited the articles in the result section.
Follow-up comment: Thank you the authors but improvement is needed. It is not necessary to cite all the 23 or 24 articles but the authors should cite and mention which 5 studies they are in the line 391.
Additional comment 1: For the table legend, make sure rearrange the abbreviations in alphabetical order.
Additional comment 2: Kindly comment the quality/risk of bias of preclinical studies as part of the limitations of study. Besides, please be specific on the line 479 (current studies), whether you were mentioning the preclinical and/or clinical studies.
Additional comment 3: For the risk of bias, do the authors categorise each articles (in vivo and clinical studies) into low, moderate or high quality?
Author Response
We thank the reviewer for the suggestions and have responded to the requests in red in the text below.
Follow-up comments:
3) Be consistent with the abbreviation, use them if necessary, define and use them consistently throughout the manuscript. Please check EV, DOCA, OA, HFD, etc.
As suggested by the reviewer, we have defined all abbreviations in the manuscript and added an abbreviations legend after each table.
- Follow-up comments: Thank you the authors for the amendment. However it is not consistent. Kindly review and remove those abbreviations that used only once. Besides, make sure define all the abbreviations and used them consistently throughout the text. Please check the Evs.
- As suggested by the review we recontroled the abbreviations and removed those that we used only once.
8) Figure 2, what does it mean by "excluded by title and abstract"? The screening should be done after remove the duplicate, not before.
We corrected Figure 2.
- Follow-up comment: Figure 2 is not corrected as stated. Why exclude 142 articles before removing the duplicates?? Please correct it.
- We corrected Figure 2 as requested by the reviewer and following the PRISMA 2020 statement.
9) Line 396: Wrong citation for PRISMA guidelines and checklist. Why not using PRISMA 2020 guidelines as there are several relevant reviews?
We corrected the reference and added the PRISMA 2020 guidelines.
- Follow-up comments (line 183): Thank you the authors for the correction. However, PRISMA 2020 is not merely a citation. The authors neeed to compliant with its guidelines and provide the 27-item checklist. Additionally, PRISMA flow diagram (Figure 2) should be updated according to PRISMA 2020. Please double check whether the authors followed the PRISMA 2020 or 2009? Why not following PRISMA 2020? Please attach the PRISMA checklist
- As suggested by the reviewer we added the 27-item checklist in the Supplementary Materials (Table 3S). Additionally, we modified Figure 2 following the PRISMA 2020 statement (Page MJ, McKenzie JE, Bossuyt PM, Boutron I, Hoffmann TC, Mulrow CD, et al. The PRISMA 2020 statement: an updated guideline for reporting systematic reviews. BMJ 2021;372:n71. doi: 10.1136/bmj.n71.)
10) What are the inclusion and exclusion criteria? Need to have a clear cut, especially on the intervention or specific therapies (line 83).
As suggested, we tried to better specified inclusion and exclusion criteria. However, for intervention or specific therapies we eliminated only articles that did not specified intervention or specific therapies used and articles in which data were not accessible or missing.
- Follow-up comments (line 232 and Figure 2): Kindly state the inclusion and exclusion criteria. Incorporate selecting criteria in Figure 2 is not suitable.
- As suggested by the reviewer we modified Figure 2 and eliminated inclusion and exclusion criteria from the Figure.
11) Table 1: Please proofread as some information is wrong, for example, use DNA sequencing for 16s-RNA.
We proofread table 1 as suggested.
- Follow-up comments: Table 2 (after renamed) is still found with several errors. DNA sequencing for 16s rDNA, not RNA. Kindly use the consistent citation/referencing style for tables.
- As suggested, we corrected and modified Table 2.
12) Table 1: Please include the respective critical appraisal score for each article. How the authors identify or judge the quality of an article is high or poor (line 357)?
We include the critical appraisal score for each article (preclinical and clinical) in the Supplementary Materials. The methodological quality of the included clinical studies was assessed following the ROBINS-I tool while the methodological quality of the included in vivo studies was done according to the SYRCLE tool.
- Follow-up comment: The authors should include the overall scoring for ROBINS and SYRCLE in the Table 2 and 3 (after renamed).
- As suggested by the reviewer we included the overall scoring for ROBINS and SYRCLE in the Table 2 and 3.
13) Table 3: Duplicate entry for PubMed. Why limit the search to 10 years? (line 386-387)
The decision to limit this systematic review to the last 10 years was influenced by several factors. Firstly, it allowed for the incorporation of up-to-date and relevant evidence in light of the significant changes that often occur in scientific knowledge and clinical practices over a decade. Additionally, more recent studies are considered more clinically relevant as they may reflect advances in therapies, diagnostic technologies, or treatment paradigms. Finally, the choice to focus on the last 10 years ensured that the included studies are in line with current research methodologies and adhere to contemporary publication standards. Furthermore, limiting the time frame has certainly helped in the synthesis of a more current body of evidence, also simplifying the systematic review process.
- Follow-up comment: The duplicate entry for PubMed (Table 1) is not corrected.
- We eliminated the duplicate entry for PubMed (Table 1).
14) Result (line 150-153), please cite the articles, not their number.
As suggested by the reviewer, we cited the articles in the result section.
- Follow-up comment: Thank you the authors but improvement is needed. It is not necessary to cite all the 23 or 24 articles but the authors should cite and mention which 5 studies they are in the line 391.
- We thank the reviewer. After the review suggestion we cited the articles in the results section. Additionally, we corrected the number that was incorrect. The studies were n=4 and not n=5.
Additional comment 1: For the table legend, make sure rearrange the abbreviations in alphabetical order.
We putted the abbreviations in alphabetical order.
Additional comment 2: Kindly comment the quality/risk of bias of preclinical studies as part of the limitations of study. Besides, please be specific on the line 479 (current studies), whether you were mentioning the preclinical and/or clinical studies.
We added quality/risk of bias of preclinical studies as part of the limitations of study and clarified the sentence “current studies”.
Additional comment 3: For the risk of bias, do the authors categorise each articles (in vivo and clinical studies) into low, moderate or high quality?
We categorized each article (in vivo and clinical studies) into low, moderate or high quality and, as suggested by the reviewer we reported these information’s in Table 2 and 3.
Reviewer 4 Report
Comments and Suggestions for Authors
The authors have addressed most of my concerns, and the manuscript has improved. However, a PROSPERO registration number (per PRISMA guidelines) should be added to the methods section before acceptance.
Figure 1. “Prepyotics”. Please correct.
Author Response
The authors have addressed most of my concerns, and the manuscript has improved. However, a PROSPERO registration number (per PRISMA guidelines) should be added to the methods section before acceptance.
We submitted our record to PROSPERO, and we are waiting for your registration number.
Figure 1. “Prepyotics”. Please correct.
We corrected the word in Figure 1.
Round 3
Reviewer 3 Report
Comments and Suggestions for Authors
Thank you to the authors for the amendment.
Follow-up comments:
9) Line 396: Wrong citation for PRISMA guidelines and checklist. Why not using PRISMA 2020 guidelines as there are several relevant reviews?
We corrected the reference and added the PRISMA 2020 guidelines.
- Follow-up comments (line 183): Thank you the authors for the correction. However, PRISMA 2020 is not merely a citation. The authors need to compliant with its guidelines and provide the 27-item checklist. Additionally, the PRISMA flow diagram (Figure 2) should be updated according to PRISMA 2020. Please double check whether the authors followed the PRISMA 2020 or 2009? Why not following PRISMA 2020? Please attach the PRISMA checklist
- As suggested by the reviewer we added the 27-item checklist in the Supplementary Materials (Table 3S). Additionally, we modified Figure 2 following the PRISMA 2020 statement (Page MJ, McKenzie JE, Bossuyt PM, Boutron I, Hoffmann TC, Mulrow CD, et al. The PRISMA 2020 statement: an updated guideline for reporting systematic reviews. BMJ 2021;372:n71. doi: 10.1136/bmj.n71.)
- Follow-up comment:
Figure 2 is not prepared according to PRISMA 2020 guidelines. Kindly mention the records from previous review. You still need to prepare that if there is no relevant review to date. How about any additional methods in literature searching? All these must be included to be compliant with PRISMA 2020.
13) Additional comment 3: For the risk of bias, do the authors categorise each articles (in vivo and clinical studies) into low, moderate or high quality?
We categorized each article (in vivo and clinical studies) into low, moderate or high quality and, as suggested by the reviewer we reported these information’s in Table 2 and 3.
Follow-up comment: There is no description of how the authors grouped them as low, moderate or high risk of bias.
14) Additional comment 4: Please explain further the registration and protocol. Are you registered with the protocol on PROSPERO? Please justify why it is not being done as you have mentioned following PRISMA guidelines.
Author Response
Follow-up comments:
9) Line 396: Wrong citation for PRISMA guidelines and checklist. Why not using PRISMA 2020 guidelines as there are several relevant reviews?
We corrected the reference and added the PRISMA 2020 guidelines.
Follow-up comments (line 183): Thank you the authors for the correction. However, PRISMA 2020 is not merely a citation. The authors need to compliant with its guidelines and provide the 27-item checklist. Additionally, the PRISMA flow diagram (Figure 2) should be updated according to PRISMA 2020. Please double check whether the authors followed the PRISMA 2020 or 2009? Why not following PRISMA 2020? Please attach the PRISMA checklist
As suggested by the reviewer we added the 27-item checklist in the Supplementary Materials (Table 3S). Additionally, we modified Figure 2 following the PRISMA 2020 statement (Page MJ, McKenzie JE, Bossuyt PM, Boutron I, Hoffmann TC, Mulrow CD, et al. The PRISMA 2020 statement: an updated guideline for reporting systematic reviews. BMJ 2021;372:n71. doi: 10.1136/bmj.n71.)
Follow-up comment:
Figure 2 is not prepared according to PRISMA 2020 guidelines. Kindly mention the records from previous review. You still need to prepare that if there is no relevant review to date. How about any additional methods in literature searching? All these must be included to be compliant with PRISMA 2020.
We thank the review for the suggestion. We modified Figure 2 and prepared it according PRISMA 2020 guidelines, as reported in:
http://prisma-statement.org/prismastatement/flowdiagram.aspx?AspxAutoDetectCookieSupport=1
Concerning additional methods for literature searching, we used ClinicalTrials.gov. However, since the records founded were all incomplete or without posted data, we evaluated these studies in the discussion section.
13) Additional comment 3: For the risk of bias, do the authors categorise each articles (in vivo and clinical studies) into low, moderate or high quality?
We categorized each article (in vivo and clinical studies) into low, moderate or high quality and, as suggested by the reviewer we reported these information’s in Table 2 and 3.
Follow-up comment: There is no description of how the authors grouped them as low, moderate or high risk of bias.
We added in the figure legends of Figure 3 and 4 a description of how the authors defined the studies as low, moderate or high risk of bias.
14) Additional comment 4: Please explain further the registration and protocol. Are you registered with the protocol on PROSPERO? Please justify why it is not being done as you have mentioned following PRISMA guidelines.
We submitted our record to PROSPERO, and we are waiting for the registration number (ID number 492912).
Round 4
Reviewer 3 Report
Comments and Suggestions for Authors
14) Additional comment 4: Please explain further the registration and protocol. Are you registered with the protocol on PROSPERO? Please justify why it is not being done as you have mentioned following PRISMA guidelines.
We submitted our record to PROSPERO, and we are waiting for the registration number (ID number 492912).
Follow-up comment: PROSPERO is not allowed post-synthesis and analysis registration. Kindly register it at OSF (https://osf.io/), please go for OST Registries and choose "Generalised systematic review registration". Lastly, please update the manuscript accordingly and state the DOI of registration
Author Response
14) Additional comment 4: Please explain further the registration and protocol. Are you registered with the protocol on PROSPERO? Please justify why it is not being done as you have mentioned following PRISMA guidelines.
We submitted our record to PROSPERO, and we are waiting for the registration number (ID number 492912).
Follow-up comment: PROSPERO is not allowed post-synthesis and analysis registration. Kindly register it at OSF (https://osf.io/), please go for OST Registries and choose "Generalised systematic review registration". Lastly, please update the manuscript accordingly and state the DOI of registration.
As suggested by the reviewer we registered our systematic review to OSF registers (DOI 10.17605/OSF.IO/5JUQC).